# Consequences of Chromosome Loss: Why Do Cells Need Each Chromosome Twice?

**DOI:** 10.3390/cells11091530

**Published:** 2022-05-03

**Authors:** Narendra Kumar Chunduri, Karen Barthel, Zuzana Storchova

**Affiliations:** 1University Medical Center Groningen, European Research Institute for the Biology of Ageing, University of Groningen, 9713 AV Groningen, The Netherlands; n.k.chunduri@umcg.nl; 2Department of molecular genetics, University of Kaiserslautern, 67663 Kaiserslautern, Germany; kbarthel@bio.uni-kl.de

**Keywords:** aneuploidy, monosomy, chromosome loss, haploinsufficiency, gene dosage, consequences of aneuploidy, chromosome loss in cancers

## Abstract

Aneuploidy is a cellular state with an unbalanced chromosome number that deviates from the usual euploid status. During evolution, elaborate cellular mechanisms have evolved to maintain the correct chromosome content over generations. The rare errors often lead to cell death, cell cycle arrest, or impaired proliferation. At the same time, aneuploidy can provide a growth advantage under selective conditions in a stressful, frequently changing environment. This is likely why aneuploidy is commonly found in cancer cells, where it correlates with malignancy, drug resistance, and poor prognosis. To understand this “aneuploidy paradox”, model systems have been established and analyzed to investigate the consequences of aneuploidy. Most of the evidence to date has been based on models with chromosomes gains, but chromosome losses and recurrent monosomies can also be found in cancer. We summarize the current models of chromosome loss and our understanding of its consequences, particularly in comparison to chromosome gains.

## 1. Introduction

Aneuploidy is a condition of unbalanced chromosome number that deviates from the usual euploid status or its multiples (Figure 1). It can result from mitotic errors, when one or more chromosomes remain unattached or improperly attached to the mitotic spindle apparatus during cell division. As a result, daughter cells either gain or lose chromosomes or a substantial part of a chromosome [1]. When such error occurs during meiosis or in early embryonic divisions, it can lead to aneuploidy of the entire organism. While some invertebrates and plants can tolerate whole-organism aneuploidy, it is embryonically lethal in most metazoans [2,3]. In rare cases, human aneuploid embryos survive to full term and can develop into adults, albeit with severe pathological consequences, such as trisomy 21, which manifests in Down syndrome [4,5]. Aneuploidy also occurs in somatic cells of healthy tissues at a low frequency [6,7]. Importantly, aneuploidy is common in cancer cells and is often associated with chromosomal instability (CIN), which is characterized by ongoing chromosome missegregation [1]. The link between aneuploidy and cancer has been well established owing to improved detection of aneuploidy, and indeed aneuploidy is one of the hallmarks of cancer [8].

The frequent occurrence of aneuploidy in cancer has stimulated general interest in understanding the consequences of aneuploidy and its role in tumorigenesis, and increased the need for suitable aneuploidy models. The first models based on the induction of chromosome missegregation provided valuable insights into aneuploidy and its contribution to tumorigenesis. However, random missegregation results in a mixed population of diploid, aneuploid, and dying cells, making it difficult to disentangle the effects of aneuploidy from confounding effects of missegregation, cell cycle arrest, or potentially occurring DNA damage. Therefore, model organisms and cell types engineered to carry one or more extra chromosomes have been developed. Their analyses indicate that although the consequences vary depending on the species, cell type, and the identity of the altered chromosome, there is a common and conserved phenotype suggesting that aneuploidy triggers a stress response commonly referred to as aneuploidy-associated stress response (ASR). ASR is characterized by altered gene expression, reduced proliferation, delayed transition from G1 to S phase, increased genomic instability, replication stress, and proteotoxic stress [9,10]. In mammals, activation of innate immune response pathways has also been observed in aneuploid cells [11,12,13]. Thus, the gain of even a single chromosome affects multiple aspects of cell physiology.

The trigger of ASR is most likely the increased expression of genes on the supernumerary chromosome, which generally scales with chromosome copy number. This leads to increased abundance of hundreds of proteins encoded on the extra chromosome, which affects protein homeostasis. These changes in protein abundance cause proteotoxic stress characterized by impaired protein folding, increased aggregation, and activation of protein degradation pathways such as autophagy and the ubiquitin-proteasome system [14,15,16,17,18]. Consequently, aneuploidy causes conserved genome-wide gene expression changes that reflect the cellular response to stressors associated with aneuploidy [19,20,21].

Most of our knowledge on aneuploidy have been obtained using model cell lines and strains that carry one or more extra chromosomes, and several recent reviews summarized the consequences of chromosome gains [9,10]. How chromosome losses affect cellular physiology and whether they have similar impact as chromosome gains remains poorly understood. Unlike trisomies, there are no autosomal monosomies compatible with viability in human embryos. Yet, whole chromosome and/or arm level losses are frequently observed in complex cancer karyotypes [22,23,24]. How can tumors survive monosomy when chromosome loss is embryonically lethal? Do the cells adapt to a loss of specific genes? Are the stresses induced by chromosome loss similar to or different from the stresses associated with chromosome gains? This review presents current evidence on the consequences of monosomy in model organisms and the role of monosomy in cancer.

## 2. Monosomy Is Detrimental in Most Organisms, but Can Be Tolerated

In most species, monosomy has adverse consequences. In mammals, autosome loss is incompatible with survival. Murine embryos with chromosome loss die significantly earlier than those with extra chromosome [5]. During post-implantation development of human embryos, trisomy 21 embryos develop similarly to diploid embryos, while monosomy 21 embryos exhibit a high rate of developmental arrest [25]. Although monosomy and trisomy arise at a similar frequency in pre-implantation embryos, no human embryonic stem cell (hESC) lines could be generated from monosomic pre-implantation embryos except for chromosome X, while several trisomic hESC lines were viable [26]. The special case of monosomy X stands out, because it is the only viable whole organism monosomy in humans, which nevertheless is associated with pathological consequences, namely the Turner syndrome (45,X). The Turner syndrome karyotype results in female phenotype owing to the presence of X and absence of chromosome Y. The mild phenotype can be explained by the specific biology of sex chromosomes in mammals. In healthy females, the gene expression of X-encoded genes is derived from only one X chromosome, while the other copy is silenced by the long non-coding RNA XIST. The RNA XIST inhibits expression of most genes encoded on the inactivated X chromosome, thus adjusting the gene dosage of XX and XY karyotypes. Only few genes (~100–130) are known to escape the XIST-mediated silencing [27]. Among them are 12 paired X-Y genes (out of 14) known to escape the XIST-mediated silencing [28]. This small number of potentially dosage sensitive genes indicates that the loss of one X chromosome does not lead to severe defects compared to an autosome loss. Similarly, embryonic loss of Y chromosome, a sex chromosome that determines the male sex results in Turner syndrome (45,X). While the Y chromosome harbors more than 70 genes, none of them are essential for viability which explains the mild phenotypes of Y chromosome loss.

Intriguingly, plants and fungi show an increased tolerance to chromosome loss, even if it is associated with lower fitness and growth defects [29]. For example, all monosomies in diploid *Saccharomyces cerevisiae* are viable, albeit with significant impairment of their fitness and reduced maintenance of genome stability [30,31]. In maize, both gains and losses of chromosomes or chromosome arms can result in morphological defects, and the severity of the phenotype depends on the chromosome identity [32]. In allohexaploid wheat, phenotypes appear to be slightly abnormal upon chromosome loss and gain in one of the three genomes that form the hexaploid genome, while nullisomics show greatly reduced vigor and fertility [33]. The increased tolerance of plants toward aneuploidy is likely due to the polyploidy occurring in many crop species, although global analysis in wheat suggested that homologous chromosomes do not specifically compensate for gene dosage changes resulting from chromosome loss [34].

Analysis of cancer karyotypes revealed that chromosome losses are more frequent than chromosome gains, but this rarely results in monosomy because many cancers simultaneously undergo a whole genome doubling [35]. Nevertheless, losses of a whole chromosome or a chromosome arm resulting in monosomy have been observed in cancers (Table 1), and some recurrent monosomic karyotypes have been associated with aggressive disease and resistance to therapy [36,37].

Recurrent chromosome changes likely arise from selection of the fittest clone, and aneuploidy facilitates this process by allowing rapid adaptation to adverse environmental conditions (e.g., [29,38,39,40,41,42]). While monosomy is less likely than polysomy to confer advantages, at least in yeast [29], several examples have been observed. In *Saccharomyces cerevisiae*, monosomy IV provides resistance toward 2-deoxyglucose, but the monosomic cells are quickly outgrown by euploids once this selection pressure is removed [43]. In *Candida albicans*, monosomy of chromosome 5 is widespread and controls susceptibility to antifugal agents [44]. The presence of recurrent monosomies in cancer, e.g., recurrent 1p deletion in neuroblastoma [45,46], 3p deletion in lung tumors [24], and the loss of 7 or 7q in myeloid leukemia [47,48,49,50] suggests that monosomy may also contribute to tumorigenesis by providing specific benefits to tumor cells, for example, through the loss of tumor suppressor genes (Table 1). Indeed, loss of 17p, where a crucial tumor suppressor gene *TP53* is located, is frequently observed in a broad spectrum of tumors. Interestingly, a mouse model of human 17p13.1 deletion, the region where human *TP53* gene is encoded, generated more aggressive tumors compared to mice with *TP53* loss alone. The aggressiveness was linked to the co-deletion of other tumor suppressors encoded in this region [51]. Loss of 9p is often associated with reduced immune cell infiltration into tumors, known as immune cell evasion [52], while loss of chromosome 10 increases cellular resistance to taxol treatment [41]. Thus, chromosome loss may contribute to tumorigenesis, but only a few recurrent monosomies have known function in tumorigenesis and a few putative effector genes have been identified. Moreover, the oncogenic effect of monosomy may result from the concerted effect of loss of multiple genes.

## 3. Model Systems for the Analysis of the Consequences of Chromosome Loss

The strong negative effect of chromosome loss on cell and organism viability have hampered our efforts to generate and maintain monosomy model systems under physiological conditions. Naturally occurring monosomic cells, for example from human embryonic material obtained from preimplantation genetic screening (PGS) samples, have been used to study the consequences of chromosome loss [26,80,81]. Although this is a valuable resource, these models have also some drawbacks: they lack isogenic controls, the findings are limited to embryonic tissues, and the cells cannot be maintained in culture for an extended period of time.

Targeted engineering of monosomy is a major challenge. In budding yeast, conditional transcription across centromeric regions leads to chromosome loss, which has been used to generate monosomic cells. These cells are, however, inherently unstable, and randomly arising euploids rapidly outgrow the population [31]. In mammalian cells, deletions of a single chromosome have also been attempted. In one approach, targeted deletion of Y-chromosome was achieved in mice using the Cre/loxP system by inserting oppositely oriented loxP sites flanking the centromeres. Cre-mediated sister chromatid recombination generated dicentric or nullicentric chromosomes, which were then eliminated during cell division [82,83]. In a second approach, a TK-NEO cassette consisting of a gene encoding a herpes simplex thymidine kinase and a gene encoding neomycin resistance was inserted into one copy of a chromosome. The TK-NEO cassette allows both positive and negative selection, as neomycin resistance selects for successful transgene insertion, while in a second step only cells that have lost the thymidine kinase gene TK survive in the presence of ganciclovir. This counterselection method was successfully used for trisomy correction in chicken DT40 cells and human Down syndrome induced pluripotent stem cells (iPSCs) [84,85]. While these approaches can in principle eliminate a chromosome, they are quite laborious, and the efficiency of obtaining clones with chromosome loss is estimated to be 1 in 10,000. Finally, CRISPR-Cas9-mediated multiple chromosome cleavage has been used to remove an entire chromosome or chromosome arm. This approach enabled efficient elimination of additional chromosomes in cancer cell lines, in mouse aneuploid embryonic stem cells with an extra human chromosome, in human iPSCs with trisomy 21, and sex chromosomes in cultured cells, embryos, and tissues in vivo [86,87,88,89]. Similarly, chromosome 3p has been removed using CRISPR-Cas9 in lung cancer cell lines with impaired p53 pathway activity [24]. Whether it is possible to remove an entire chromosome in non-transformed diploid cells using these approaches remains to be tested.

An alternative strategy was used to derive monosomic human cells that lost a chromosome due to chromosome missegregation in mitosis. Strikingly, the monosomic cells survived only when the p53 pathway was inactivated either by *TP53* knock out or by expression of an shRNA targeting p53 [90,91,92](preprint). Moreover, reintroduction of functional p53 leads to activation of the p53 pathway, manifested by the expression of p53 targets and subsequent loss of p53-positive monosomic cells from the population [91]. No constitutive autosome deletions have been established from diploid non-cancerous, p53 positive cells.

Loss of *TP53* does not lead to aneuploidy per se [93], but it does allow the cells with aneuploid karyotype to proliferate [18,90,94,95]. *TP53* is the most frequently mutated gene in cancers, and aneuploidy closely correlates with impaired p53 activity [94]. This association with p53 defects may be even stronger for monosomy. Recent evidence suggests that *TP53*-mutated MDS (myelodysplastic syndrome) and leukemia patients are enriched in chromosome 5(q) deletion and monosomy 7 [96]. Consistently, analysis of pan-cancer TCGA data revealed that tumors with monosomy display more often *TP53* pathway mutations compared to diploid and polysomic tumors [91]. This highlights the role of p53 in viability of monosomies.

## 4. Reduced Proliferation and Impaired Genomic Stability Due to Monosomy

Monosomy has a detrimental effect on cellular growth, and human monosomic cells proliferate slower and display reduced growth on soft agar, even in the absence of *TP53* [91]. Interestingly, the proliferation defect in monosomies correlates with the number of open reading frames encoded on the lost chromosome [31,91,92] (preprint). The deleterious defects in monosomy likely arise from gene haploinsufficiency (HI), or, alternatively, a loss of an allele may unmask existing recessive mutations. Of note, the cell line with monosomy X also proliferates slower than the diploid control, probably due to the loss of the transcribed genes on the inactivated chromosome X [91].

Another consequence of chromosome loss is impaired genomic stability. In budding yeast, chromosome loss was associated with prolonged cell cycle and increased DNA damage [31]. Human monosomic cells showed increased occurrence of micronuclei, anaphase bridges, and γH2AX marker of DNA damage [91]. Single-cell RNA sequencing analysis of hematopoietic stem cells from patients with myeloid malignancies with monosomy 7 showed down-regulation of genes involved in the maintenance of DNA stability [97]. In contrast, a study comparing genomic instability in monosomy and trisomy cells revealed that chromosome losses do not impair the genomic stability, while chromosome gains do [92](preprint). This discrepancy could possibly be attributed to both monosome identity as well as to the confounding chromosomal gains in some of the monosomic cell lines. Future studies of several different chromosomes are needed to clarify the impact of chromosome loss on genome stability in mammalian cells.

It is plausible to hypothesize that genomic instability in monosomic cells results from haploinsufficiency of genes responsible for genome maintenance. Genes such as PTEN (encoded on chromosome 10) [98], RB [99], BRCA2 (encoded on chromosome 13) [100] induce genomic instability when expressed at reduced levels. Experiments in which these proteins are rescued in monosomies should clarify whether the observed genomic instability in monosomies is due to their reduced expression.

## 5. Loss of a Chromosome Leads to a Reduced Expression of the Monosomic Genes

Loss of a chromosome affects both mRNA and protein abundance of all genes encoded on the missing chromosome in all organisms studied to date. When diploid cells lose a chromosome, the expected expression level should be half of the diploid expression level (0.5-fold change, or −1 log2FC; Figure 2). Indeed, analysis of the transcriptional consequences of chromosome loss in genetically distinct diploid and aneuploid blastocysts revealed reduced expression of genes encoded on the monosomes [80]. Similarly, significant downregulation of mRNA abundance was found for 64% of genes encoded on chromosome 3p following the deletion of this chromosome arm in human cells [24]. Importantly, expression does not always scale with the gene copy numbers. In a monosomy 5 strain of *Candida albicans*, nearly 40% of transcripts encoded on chromosome 5 were expressed at the diploid level [101]. Transcriptome analysis of 16 aneuploid maize lines with varied dosage of multiple chromosomal segments (monosomy to tetrasomy) showed a broad range of gene-dosage effects, from the mRNA abundance correlating with the gene dosage to dosage compensation [102]. Dosage compensation at the transcriptional level was also observed upon a loss of large fragments of sex chromosomes and autosomes in *Drosophila melanogaster*, with log2FC ranging from −0.9 to −0.52, depending on the individual monosomic region [103,104,105]. Interestingly, the compensation was anti-correlated with the number of ORFs encoded on each chromosome [105].

Analysis of non-transformed isogenic human cell lines with monosomies of five different chromosomes also revealed transcriptional dosage compensation toward diploid levels in a range from −0.52 to −0.74 of log2FC [91]. Here, compensation was greater when the number of genes encoded on the lost chromosome was low, in contrast to observations in *Drosophila.*

Moreover, proteome analysis showed that dosage compensation was even stronger, with median expression of proteins encoded on a monosomic chromosome ranging from −0.26 to −0.37 log2FC of diploid levels [91]. Thus, post-transcriptional mechanisms strongly contribute to compensate for the altered gene dosage.

Buffering the effects of chromosome copy number changes is well documented in cells with additional chromosomes [14,106,107]. Recent comparative analyses of genomes, transcriptomes, and proteomes have shown that dosage compensation is also a widespread feature in aneuploid human cancers [108,109] (both preprint). Interestingly, the extent of compensation and whether compensation occurs at the mRNA or protein level depends on tissue type. For example, lung cancer has low levels of compensation, whereas mRNA level compensation is dominant in tumors of colon, breast, ovarian, and renal origin [109]. The prevalence of dosage compensation suggests that aneuploid cancer cells benefit from mechanisms that provide substantial mitigation of the gene expression changes caused by numerical chromosomal aberrations.

Several mechanisms are conceivable to buffer the effects of monosomy on gene expression: increased transcription of the remaining chromosome, increased mRNA stability, increased translation of proteins encoded on monosomes, or reduced protein degradation. Recent studies have shown that mRNA levels determine protein abundance under physiological conditions [110,111], but during stress protein expression is strongly regulated by post-transcriptional mechanisms [112,113]. Thus, gene expression control is determined by the cellular physiological state, and the monosomy-induced stresses influence the extent of correlation between the gene expression and chromosome copy number. In addition, global analyses of the proteome suggest that often more mRNA and protein are produced than required for cellular processes and become subsequently trimmed to optimal abundance [114]. This observation could explain the efficient gene dosage compensation and lack of proteotoxic stress in monosomic cells.

## 6. Genome-Wide Expression Changes in Response to Chromosome Loss

Although only limited amount of data is available, it appears that chromosome loss affects not only the expression of the specific encoded genes, but also gene expression genome-wide. Genome-wide changes were observed in human aneuploid blastocysts where loss of a chromosome altered the expression of genes involved in essential cellular processes such as cell cycle regulation, DNA replication, metabolism, mitochondria, ribosome biogenesis, and translation [80]. Similar changes were observed in transcriptome of cancerous cells lines lacking 3p [24]. Recent transcriptome and proteome analysis in human non-cancerous cell lines lacking chromosome 10, 13, 18, 19p, or X showed a broad spectrum of chromosome-specific gene expression changes. For example, pathways related to the response to interferon and major histocompatibility complex (MHC) were specifically downregulated in the cell line lacking chromosome 13, but not in other monosomic cell lines. These deregulations could be due to the loss of specific genes encoded on individual chromosomes. For example, chromosome arm 19p encodes several NADH dehydrogenases genes (*NDUFA11*, *NDUFA13*, *NDUFA3*, *NDUFA*, *NDUFB7,* and *NDUFS7*). Haploinsufficiency of these genes likely leads to downregulation of mitochondrial respiratory chain and oxidative phosphorylation pathways, which were specifically observed in cell line with monosomy 19p [91]. The only differential pathway regulation common to all monosomic cell lines was consistent down-regulation of genes related to ribosome biogenesis, cytosolic large and small ribosomal subunits, and translation.

Functional follow-up analysis revealed that translation and rRNA abundance are reduced in monosomic cells, consistent with the pathway changes [91]. Analysis of Cancer Cell Line Encyclopaedia (CCLE) datasets confirmed that metabolic pathways related to ribosomes and translation were also downregulated in monosomic cancer cell lines, supporting the notion that chromosome loss specifically affects these metabolic pathways. However, ribosomes and translation are downregulated in response to many different stressors and have also been observed in cells with additional chromosomes [20]. One possibility is that proteotoxic stress and unfolded protein response (UPR) triggered by overexpression of genes due to chromosome gain can activate the integrated stress response (ISR, environmental stress response, ESR, in yeast), which in turn impairs ribosome biogenesis and translation. Indeed, this has been observed in budding yeasts with extra chromosomes [21], as well as in murine models of trisomy 21, where a translation deficiency in brain cells could be rescued by inactivating the integrated stress response [115]. Strikingly, monosomic cells do not show proteotoxic stress, UPR, and ISR transcriptional signature [91,92] (preprint).

An alternative explanation could be that the haploinsufficiency of ribosomal subunit genes (ribosomal protein genes, RPGs) leads to downregulation of ribosome biogenesis and translation in monosomic cells. RPGs are distributed throughout the genome, such that every chromosome except for chromosomes 7 and 21 carries at least one RPG [116]. Additionally, the human genome contains ∼400 copies of a tandemly arrayed 43-kb rDNA unit on chromosomes 1, 13, 14, 15, 21, and 22, making chromosome 7 the only one that does not encode any ribosomal subunit or rRNA. Several ribosomal protein genes are encoded on chromosome X, including RPS4X, which is one of the few genes that escape the XIST-mediated dosage compensation and thus is actively transcribed even on the silenced X chromosome [27]. Accordingly, the cell line with monosomy X displayed defects in protein translation, similar to cell lines lacking one copy of autosomes [91]. Additionally, RPG loss activates the tumor suppressor protein p53 in mammals. Ribosome biogenesis stress induced by decreased RPG expression leads to accumulation of p53 due to sequestration of MDM2 by free ribosomal subunit proteins [115] and affects cellular viability (Figure 3A). Thus, ribosomal haploinsufficiency may also explain the incompatibility of monosomy with functional p53 pathway observed in human cells.

Ribosomal haploinsufficiency has long been associated with several pathological conditions known as ribosomopathies. Diamond-Blackfan anemia is a well-studied ribosomopathy caused by mutations or deletion of RP genes such as *RPS7*, *RPS10*, *RPS17*, *RPS19*, *RPS24*, *RPS26*, *RPS29*, *RPL5*, *RPL11*, *RPL26*, and *RPL35A* [117]. The pathology of 5q syndrome, an independent subtype of myelodysplastic syndrome, is specifically associated with haploinsufficiency of *RPS14*, which is encoded on chromosome 5 [118]. However, cancer cells often upregulate ribosomes and translation to support their increased proliferation, and this is true also for tumors with monosomy, as found by analysis of the transcriptomes in the TCGA database (NKC, unpublished data). This suggests that monosomic cancer cells have evolved to upregulate the ribosomes and translation despite chromosome loss and RP haploinsufficiency. Therefore, disruption of ribosome biogenesis may be an effective anticancer treatment [119], and tumors with monosomies might be particularly sensitive to this treatment.

## 7. Haploinsufficiency as the Main Detrimental Consequence of Chromosome Loss

The adverse effects of whole chromosomal monosomy and segmental deletions were previously attributed to haploinsufficiency of genes encoded on the monosomic chromosome. In addition to chromosome loss, other defects can lead to decreased gene expression, such as loss-of-function mutation, reduced mRNA and protein stability, or impaired translational control (Figure 3B). High-throughput screens in budding yeast identified only about 3% (180 genes) of the genes in the entire genome as haploinsufficient, but this number increased up to 20% when yeasts were grown in limited nutrient conditions [120]. Similarly, analysis of heterozygous yeast deletion strains revealed haploinsufficient phenotypes for approximately half of 1112 essential genes under normal growth conditions; 40% of genes that did not show a phenotype under normal growth conditions exhibited haploinsufficiency under severe growth conditions [121]. These results suggest that stress conditions induce haploinsufficiency by increasing the requirements for full functionality of certain physiological processes. In humans, approximately 300 genes appear to be haploinsufficient and have been associated with pathological conditions such as cancer, developmental and neurological disorders [122,123], but computational analysis predicts that the number of haploinsufficient genes could be much higher [124]. The haploinsufficient genes are often involved in essential cellular processes such as transcription, translation, cell cycle, and development, which could also explain the embryonic lethality of chromosome loss.

Two models have been proposed to explain haploinsufficiency (Figure 3C). The first, the *insufficient amount hypothesis*, suggests that loss of an allele leads to reduced protein abundance, thereby affecting the downstream functions of the protein [125]. A classic example for this hypothesis is transcription factors, because their reduced amount subsequently reduces the expression of their target genes, thereby strongly affecting cellular processes. For example, haploinsufficiency of a zinc finger transcription factor GATA-4 caused by deletion of chromosome region 8p23.1 contributes to congenital heart diseases [126]. The second model is referred to as the *dosage balance hypothesis*, wherein the haploinsufficiency is caused by perturbations in the stoichiometry of multiprotein complexes [127]. A classic example of the dosage balance hypothesis is the haploinsufficiency of ribosomal subunit genes. Loss of even one ribosomal gene significantly affects ribosomal stoichiometry and impairs cellular fitness [128]. Interestingly, the *dosage balance hypothesis* suggests that haploinsufficient genes affect cellular fitness when their gene copy number is altered. The narrow range for optimal activity also explains why haploinsufficient genes are not eliminated during evolution, and their expression is not increasing [129].

Interestingly, haploinsufficiency is often associated with cancers, largely due to a loss of tumor suppressor genes. Haploinsufficiency of tumor suppressor genes encoding cell cycle regulators (p27KIP1, p53, p21, RB, and DMP1), signaling molecules (PTEN, SMAD4, and LKB1), and genes maintaining genomic stability (MSH2, MAD2, BRCA1 and 2) has been linked to cancer [130]. In addition to loss of single genes, deletions of chromosome arms such as 5q, 7q, and 8q in cancers result in loss of multiple tumor suppressor genes, termed compound haploinsufficiency [49,131,132]. While the effects of haploinsufficiency varies from gene to gene, it is clear that some gene losses impair cellular fitness, while others contribute to tumor development.

The establishment of vertebrate somatic haploid cell lines as well as the existence of viable haploid organisms, such as in unicellular eukaryotes [133], brings forward the question why loss of one chromosome leads to more severe phenotypes compared to haploids where the whole genome is present in a single copy. While haploid cells, particularly from vertebrates, show some phenotypic changes, such a reduced proliferation and increased genomic instability compared to diploids [134], the recent data suggest that it is the relative dosage of chromosomes which strongly determines the fitness of the organism. Loss of single chromosome leads to reduced dosage of genes encoded on that chromosome, which may directly affect specific pathways and additionally disturb the stoichiometry of protein complexes. In contrast, the relative dose of chromosomes remains unchanged in haploid cells compared to diploids, and therefore the stoichiometry is unaltered.

## 8. Cellular Response to Monosomy Differs from the Response to Trisomy

While the cellular response to monosomy and trisomy looks similar at a first glance—both gains and losses result in decreased proliferation and many physiological changes—it is noteworthy that only a few phenotypes are common, and the genome wide gene expression changes in cells with gain and loss of a chromosome are rather different (Figure 4). The cellular response to chromosome gains is thought to be caused by the low but chronic overexpression of several hundreds of genes on the extra chromosomes, leading to proteotoxic stress [9,10]. Protein homeostasis is affected by increased protein abundance and by imbalanced stoichiometry of subunits of multimolecular complexes. In contrast, chromosome loss does not saturate the protein folding machinery, as there is no excess proteins to be folded. Although the imbalanced stoichiometry due to chromosome loss could potentially cause proteotoxic stress, monosomies were not sensitive to 17-AAG, an inhibitor of a protein folding factor HSP90. Further, monosomy does not affect autophagy, in contrast to cellular response upon chromosome gains [91]. Together, these findings suggest that monosomic cells do not suffer from proteotoxic stress. In addition, presence of even one extra chromosome causes replication stress and reduced expression of DNA replication-dependent proteins [11,18,135,136]. However, chromosome loss does not lead to replication stress, and expression of replication stress markers, such as phosphorylation of RPA32 and CHK1 proteins remained unchanged in monosomies compared to diploids [91,92] (preprint). Additionally, pathway changes previously identified in cells with extra chromosome were not observed in monosomic cells [91]. Thus, it appears that chromosome loss elicits a strikingly different response than chromosome gain. However, there are limitations that make a direct comparison of the currently available monosomic and tri-/tetrasomic model systems difficult. First, the altered chromosomes are not identical, so the chromosome-specific effects can occlude the conclusions. Second, the findings to date have been based on few karyotypes, and the genetic background of the current model systems does not allow direct comparison, for example, due to differential expression of p53. Future studies should aim to obtain isogenic cell lines for identical chromosome losses and/or gains to directly investigate the differences and similarities in cellular response to chromosome gains and losses.

## 9. Conclusions and Perspective

Recent success in manipulating the karyotypes of eukaryotic cells allowed comparison of isogenic cell lines with specific chromosome changes and demonstrated that the common cellular response to chromosome gains differs from that upon chromosome losses. Often, chromosome-specific phenotypes seem to dominate the cellular response to monosomy [52,91]. Only reduced proliferation and decreased translation, and possibly impaired genome stability appear to be general consequences of monosomy. This may be caused by haploinsufficiency of DNA-repair genes or of genes encoding ribosomal subunits located on the specific missing chromosomes. Future experiments should test this hypothesis by overexpressing the relevant genes, although this may be difficult, because haploinsufficient genes are highly sensitive to expression levels and their products are fully functional only within a narrow abundance range. Another open question remains how general are the findings that were so far obtained in the few available monosomic cell lines. The observed downregulation of ribosomes and translation in monosomic cells again points out the “aneuploidy paradox”. While monosomy leads to a wide range of cellular defects, it is frequent in cancer and often occurs recurrently, indicating that it might bring important fitness advantages to tumors. However, monosomic cancer cells have to adapt to the defects caused by chromosome loss, and evolve, for example, to upregulate the ribosome gene expression and translation despite chromosome loss. This raises a variety of questions. Is the ribosomal haploinsufficiency indeed the main consequence of monosomy? How does ribosomal haploinsuffiency contribute to tumorigenesis? And how do tumor cells with monosomy achieve higher expression of genes involved in ribosomal and translational pathways? Identification of factors leading to an increased ribosome synthesis and translation might provide new therapeutic targets for cancer treatment. Finally, the link between RPG haploinsufficiency and p53 function might play a key role in the viability of monosomies. Overexpression of ribosomal subunits might therefore provide insight into a potential p53 stabilization and activation via MDM2 sequestration. Alternately it should be addressed, if p53 is activated by DNA damage and if this phenotype results from chromosome-specific or general responses to chromosome loss. Finally, future studies should also aim to obtain a broader spectra of model cell lines with various karyotypic changes to test the generality of recently obtained findings.

## Figures and Tables

**Figure 1 cells-11-01530-f001:**
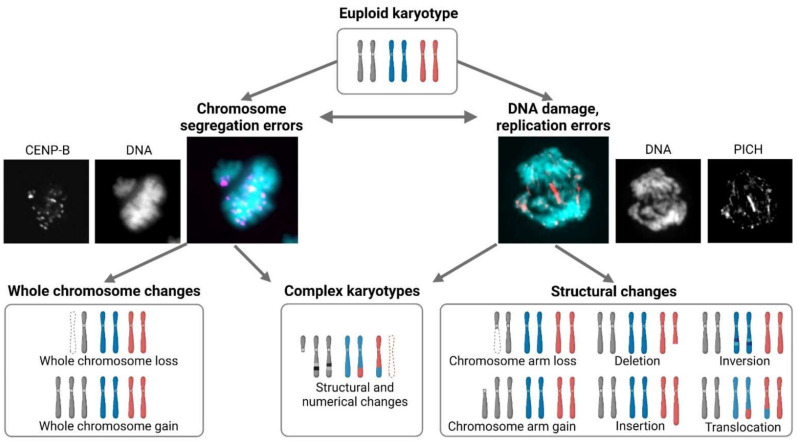
Karyotypic changes and their origins. Eukaryotic cells with euploid karyotype contain usually a diploid chromosome set. Chromosome segregation errors during mitosis result usually in whole chromosome alterations. Chromosome missegregation can occur through various defects in centromere, kinetochore, and spindle proteins. Unattached and lagging chromosomes can be detected by microscopy. DNA damage or replication stress can result in unrepaired or underreplicated DNA, which manifests itself as defect during anaphase as chromatin and/or ultrafine bridges that are visualized by immunostaining of associated proteins (such as PICH). Lack of repair and erroneous resolution of these bridges leads to chromosome breakage and subsequent losses or gains, insertions, deletions, inversions, or translocations. It should be noted that missegregation can also result in DNA damage and vice versa. DNA repair and replication defects may increase the occurrence of segregation errors. Karyotypes in cancer are frequently diverse, displaying combinations of structural and numerical changes. DNA staining (DAPI; cyan), immunostaining of a centromeric protein CENP-B (magenta), and the helicase PICH (red).

**Figure 2 cells-11-01530-f002:**
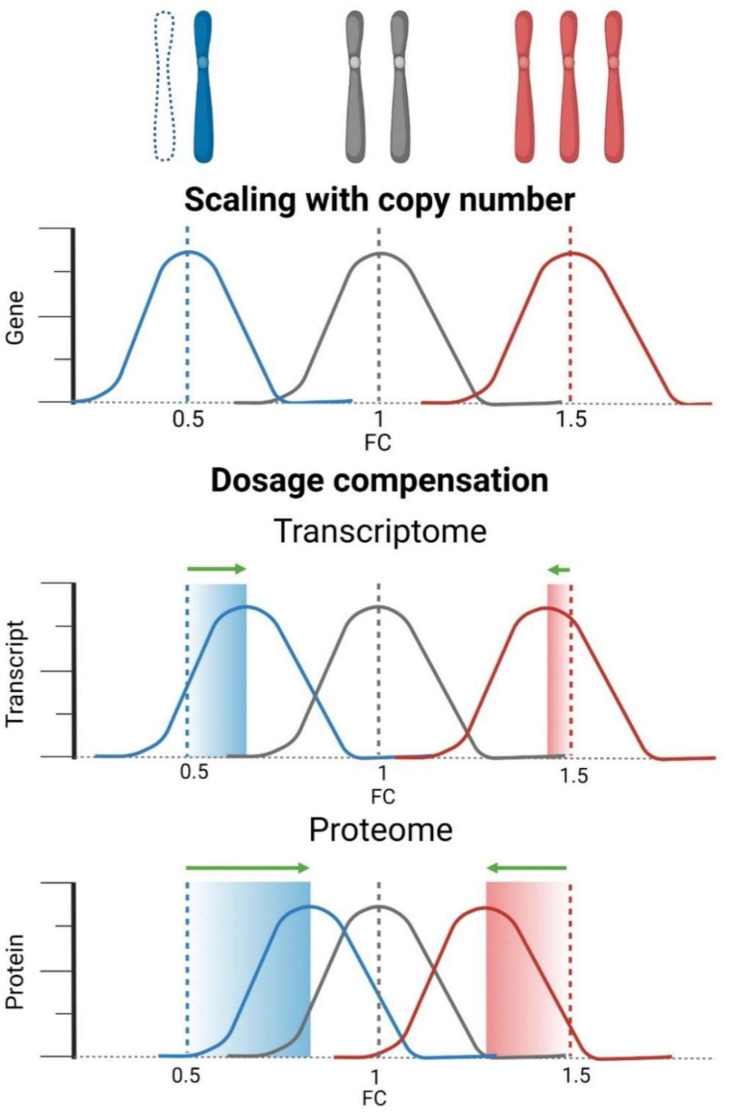
Gene dosage compensation on transcriptome and proteome level. Monosomy (blue) results in a gene dosage reduction, which is manifested in 50% reduction compared to the diploid status. This can also be expressed as the fold change (FC); the expected median (dashed line) is 0.5 compared to the diploid status 1 (grey), or log2FC = −1 compared to the diploid status log2FC = 0 (not depicted). In reverse, a chromosome gain (red) results in a 50% increase compared to diploids, which is a FC of 1.5, and log2FC = ~0.58. If gene expression reflects the copy number alterations, the same changes will be detected on transcriptome and proteome level. If gene expression becomes compensated toward diploidy on the transcriptome level and proteome level, the calculated FC compared to diploid status changes. Strikingly, the dosage compensation is often stronger at the proteome level than at the transcriptome level, and stronger upon chromosome loss than upon chromosome gain.

**Figure 3 cells-11-01530-f003:**
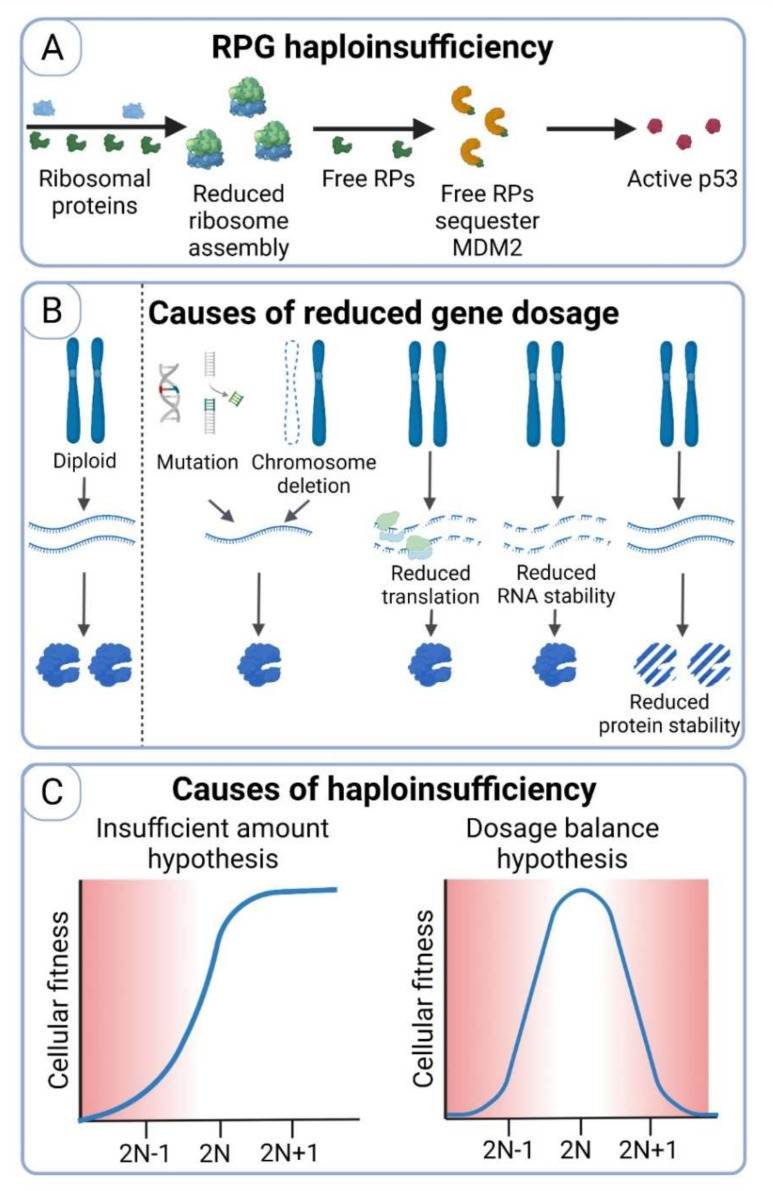
Causes and consequences of haploinsufficiency. (**A**) Reduced amount of a single ribosomal protein leads to impaired ribosome assembly. This impairs proteostasis and leads to reduced translation. Additionally, certain free ribosomal proteins interact with MDM2, the E3 ligase that ensures degradation of p53 in absence of cellular stresses. If MDM2 is bound to RPs, p53 is stabilized. (**B**) Gene dosage reduction occurs through multiple processes. The DNA level can be affected by loss-of-function mutation, segmental, or whole chromosome deletion. Reduced mRNA stability and translation, as well as reduced protein stability may also decrease the abundance. (**C**) Illustration of the so-called *insufficient amount hypothesis*, which proposes that a reduced protein amount is not sufficient for wild type functions. In this case, an increased amount of the respective protein does not impair cellular fitness. The *dosage balance hypothesis* suggests that haploinsufficiency results in an imbalanced protein stoichiometry. In this case, both decreased and increased protein amount leads to unbalanced protein stoichiometry.

**Figure 4 cells-11-01530-f004:**
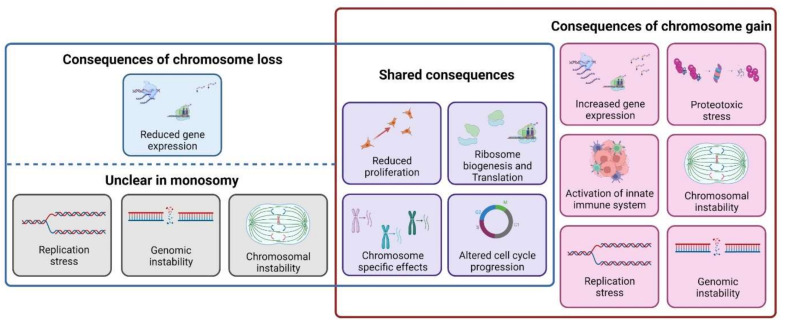
Consequences of aneuploidy. Alterations from the euploid karyotype trigger several cellular responses that are shared upon chromosome loss and gain, including reduced proliferation, impaired ribosome biogenesis and translation as well as chromosome-specific effects (purple panels). However, some consequences, such as increased gene expression or proteotoxic stress are unique to chromosome gains (red). In contrast, reduced gene expression is the only feature so far detected solely in monosomies (blue). Some physiological effects commonly detected upon chromosome gain might also be found upon chromosome loss, although limited data on monosomy hinder a clear assertion on these phenotypes (grey).

**Table 1 cells-11-01530-t001:** Examples of recurrent monosomies in cancer. The list of identified recurrent monosomies associated with cancer, and the identified putative genes whose loss contributes to tumorigenesis in respective monosomies. The frequencies of the chromosome or chromosome arm loss differ in various studies, and depend on the tissue and grade.

Affected Chromosome	Cancer Type	Putative Effector Gene	Frequency of Deletion	Reference
1p	Neuroblastoma	*MYCN*	5–52% [53]	[45,54]
1p, 19q	OligodendrogliomaAstrocytic gliomas, Glioblastoma		70% [55] 2–25%; 4–12% [56]	[57,58]
3p	Lung carcinoma, Lung squamous cell carcinoma	Multiple, reviewed in: [59]	80% [24]	[60,61]
3	Uveal melanoma	*BAP1*	50% [62]	[63,64]
5/5q	Myeloblastic syndrome	*EGR1*, *APC*, *DIAPH1*, *NPM1*	n/a	[65,66]
7/7q	Myeloblastic syndrome, Myeloid leukemia	*CUX1* [67], *MLL3* [50]	12–70%	[65,68,69]
8p	Prostate carcinoma Renal clear cell carcinoma		~13%	[70,71,72]
9	Bladder cancer	*ARF*, *TGF**β*		[73,74]
13	High-grade glioma	*BRCA2*	90%	[75]
17p	Multiple myeloma, Lymphocytic leukemia, Colorectal cancer, Medulloblastoma	*TP53*, *BRCA1*	n/a	[76]
22	Meningioma	*NF2*	40–75% [77]	[78,79]

n/a—data not available.

## Data Availability

Not applicable.

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
