# Peer review of "Consequences of Chromosome Loss: Why Do Cells Need Each Chromosome Twice?"

_cells, 2022, doi:10.3390/cells11091530_

Round 1
Reviewer 1 Report
This manuscript contains a comprehensive and authoritative review of a less investigated aspect of aneuploidy, the causes and consequences of chromosome loss and comparisons to the more studied chromosome gain-type aneuploidies. As such the work will be of significant interest to researchers studying chromosome instability. As is, the manuscript is well written. However, I offer the following suggestions.
It is of useful note that lower eukaryotes such as yeast and even plants are discussed to some extent. It would be of great interest to expand the information about plants since most mammalian researchers know very little in this area, but it is understood that some space limitations must be met.
Perhaps the major lack that I see with the manuscript is the exclusion of comparison of monosomies with haploid situations that might be useful in underpinning some of the considerations of elements such as the dosage balance hypothesis, which while discussed toward the end of the manuscript, does not in my opinion receive the attention it deserves. The obvious example is yeast where haploid cells are physiologically normal. But other examples exist, among whole animals such as amphibians as well as derived mammalian cell lines.
Other criticisms are more along the lines of personal opinions. For example, the authors discussion, highlighted in Figure 1, touting segregation errors as the source of numerical chromosome aneuploidies and DNA damage/replication errors as the source of structural chromosome defects is, I think, a bit too simplistic.
There are a few other examples where there probably should be some measured uncertainty about conclusions in the wording. For example on line 209, discussion monosomies of X, the wording suggests a firm conclusion that loss transcribed genes on the inactivated X are responsible for slowed growth, which goes a bit beyond the unequivocally demonstrated data.
The manuscript contains a significant number of self-references. This is not entirely unexpected since the home lab has contributed significantly in this area. However, one reference, number 90, referred to numerous times in the manuscript is from the laboratory but the author list in the citation is truncated and does not list all the authors, perhaps a defect in the citation application used. This problem should be corrected in all the references.
Generally, I found the figures informative at conveying the important concepts with the exception of Fig 3A where I found the right side pictographs contain little information and the detail is conveyed by words, a definition of a faulty figure. (There is a typographical error in figure 3B.)
My last suggestion would be to minimize the use of acronyms. Acronyms hamper readability, particularly for individuals outside the field. With some exceptions, e.g. DNA, it is best to avoid them when possible. In the current work, I take as the easiest to eliminate the acronym HI, which appears to sub for the single word haploinsufficiency. The acronym is used inconsistently and apparently at random compared to the whole word. A list of essential acronyms at the beginning of the manuscript would be useful.
Reviewer 2 Report
Dear authors, the paper is very interesting and can be accepted for publication in Cells. I really enjoyed reading this manuscript and I do not have any change to request.
Author Response
Reviewer 2
Dear authors, the paper is very interesting and can be accepted for publication in Cells. I really enjoyed reading this manuscript and I do not have any change to request.
>>We thank the reviewer for the positive feedback.
Reviewer 3 Report
This review paper was very well written and focused on the consequences of chromosome loss, which is one of the hottest topics in cancer and developmental research fields. I really enjoyed reading this review and found multiple valuable information. Authors covered from previous papers to the latest bio-archive papers. I only have few minor comments. Minor concerns were listed below.
Major concerns:
- Authors covered well about the chromosome loss of autosomes and also some about X-chromosome in female case. I understood why since the review focused on paired chromosomes. However, it would be more informative and helpful if the author can briefly cover Y-chromosome loss in this review.
- I do believe this is good to cite the latest bio-archive papers, and those works were performed by highly trustable researchers. However, I think you should aware readers that those are bi-archive studies, which do not process the peer-review yet and may significantly change their results and conclusions.
Reviewer 4 Report
Chunduri et al. summarize recent findings on the effect of chromosome loss. Chromosome loss is observed in cancer cells as well as chromosome gain. However, not much is known about the impact of chromosome loss on cells compared to chromosome gain. This review summarizes the effect of chromosome loss.
Major points:
- How do the authors think the “aneuploidy paradox (introduced in the abstract)” could be explained or solved?
- I suggest clarifying and discussing a few essential but unsolved questions in this field.
Minor points:
- Line 111. ‘…but difficult to maintain’. ‘maintain’ what?
- Line 117-119. ‘The increased tolerance….’ This sentence is hard to understand. Please rewrite this.
- Line 138-141. ‘However, analysis in human cells….’ This sentence is hard to understand. Please rewrite this.
- Table 1. The relationship between ‘Affected chromosome’ and ‘Cancer type’ is unclear. For example, ‘17p’ and ‘kaemia, colorectal cancer, medullo-blastoma’
- Line 208. ‘haploinsufficiency (HI)’ Once the authors started using the abbreviation, it is better to keep using it. For example, Line 225 ‘haploinsufficiency’
- Line 289. ‘lack of proteotoxic stress’ Can the authors explain the proteotoxic stress.
- Lines 334-335. ‘Chromosome X carries besides RPL10, RPL36A and RPL39 also RPS4X’ I cannot understand why ‘besides’ and ‘also’.
- Line 342. ‘… by unbound ribosomal proteins’ ‘unbound’ from what?
- Figure 3. It is better to make the order of panels (A, B, and C) consistent in the figure and the main text.
- Figure 4 legend. Indicate which is the explanation of the purple panels.
